# Exploring School Staff Perceptions Relating to Animals and Their Involvement in Interventions to Support Mental Health

**DOI:** 10.3390/ijerph19127126

**Published:** 2022-06-10

**Authors:** Rhoda A. Leos, Paula M. Cuccaro, John R. Herbold, Belinda F. Hernandez

**Affiliations:** School of Public Health, University of Texas Health Science Center at Houston, Houston, TX 77030, USA; paula.m.cuccaro@uth.tmc.edu (P.M.C.); john.r.herbold@uth.tmc.edu (J.R.H.); belinda.hernandez@uth.tmc.edu (B.F.H.)

**Keywords:** animal-assisted interventions, mental health, childhood trauma, school interventions

## Abstract

Given the growing awareness of the health benefits of human–animal interactions, the use of animal-assisted interventions (AAIs) in educational settings has increased over the years. While many school districts are now considering or utilizing AAIs, the literature investigating AAI-related perceptions among school stakeholders is limited with previous studies focusing on evaluating specific programs. To address this gap, a qualitative exploratory study was conducted using semi-structured interviews with school staff in the San Antonio, TX community. A total of 11 interviews were completed with staff serving preschool and elementary school age children. Data collected from interviews were analyzed using thematic analysis. Findings demonstrated that participating staff had some knowledge of the potential benefits of human–animal interactions and perceived the involvement of animals in interventions to be beneficial to children’s emotions and social-emotional skills. While perceptions of AAIs were generally positive, concerns around children’s safety and well-being were expressed as potential barriers in the adoption of AAIs in schools. These findings are preliminary and provide a segue to future research that can help expand our understanding of how school staff perceive AAIs, their impact on children’s mental health, their compatibility with school values, and their advantage relative to other interventions.

## 1. Introduction

Mental health and early trauma have become issues of priority for many communities. While children of all ages can experience trauma, young children are of particular concern. Research has shown that child abuse and neglect are highest among children under three years of age and that children aged 5 years and younger are more likely to experience domestic violence and substance abuse among parents [1] (p. 360). Young children who experience trauma are more likely to encounter a wide array of social-emotional challenges including difficulty controlling their temper and making friends, as well as difficulties in planning, problem-solving, and self-regulating responses [2,3]. Given these consequences and the long-term health effects of childhood trauma, including mental illness, efforts to create and sustain safe environments and nurturing relationships are essential [4].

Cities and communities such as San Antonio have taken action to address children’s mental health and expand trauma-informed efforts. In recent years, the City of San Antonio Metropolitan Health District has launched several trauma-informed initiatives, including the implementation of the “Triple P” evidence-based positive parenting program and the collaboration with partners for the South Texas Trauma-Informed Care Consortium [5,6]. School districts have also implemented programs and initiatives, including the “Handle with Care” program where police provide notice to schools when a child is involved in a traumatic event [7]. As San Antonio continues expanding efforts to support mental health among children, a unique group of interventions has surfaced and gained interest in the school community. While these interventions may share similar goals with other programs and interventions, what makes them unique is their involvement of animals.

Animal-assisted interventions (AAIs) is an umbrella term that includes animal-assisted therapy, animal-assisted crisis response, and other structured interventions that “intentionally incorporate animals in health, education, and human service for the purpose of therapeutic gains and improved health and wellness” [8], (para. 1). These interventions have also been broadly defined as “the use of animals as part of educational and therapeutic interventions for humans.” [9]. Since Friedman and colleagues published their study in 1980, demonstrating the health benefits of pet ownership, a growing amount of research has continued to showcase the physiological and psychological effects of human–animal interactions. Such effects include decreases in blood pressure, symptoms linked depression and anxiety, and feelings of loneliness [10,11]. Furthermore, the idea of incorporating animals in interventions aimed at addressing trauma is not a new one. Research has shown animals may be particularly effective at “reworking patterns of insecurity” among individuals that have experienced trauma and who mistrust others [12] (p. 1). Meta-analyses have found the involvement of animals to be an effective part of treatment for trauma-exposed individuals and reducing post-traumatic stress symptoms [13,14]. In addition, researchers have highlighted the role that companion animals can play in managing uncertainty during the COVID-19 pandemic and preventing post-traumatic stress symptoms [15].

In relation to AAIs implemented in educational settings or with school children, studies have found positive outcomes among participating children, including reductions in aggression, improved emotional stability, increased positive attitudes toward school, and enhanced learning on topics such as respect and empathy [9]. Specific to children’s mental health, a 2019 pilot study found that an AAI program helped reduce symptoms related to post-traumatic stress disorder among children who had been exposed to domestic violence [16]. A 2018 qualitative study with children that had experienced parental substance use found animal-assisted therapy to provide a safe and supportive environment and help improve social behaviors [17]. In 2020, a study found that integrating an AAI (specifically a humane education curriculum) alongside a behavioral health treatment program reduced symptoms and improved mood for children with certain psychiatric issues [18]. While these findings showcase AAIs and human–animal interactions as protective factors, we acknowledge that this area of research is still developing, and more research is needed to better understand the impact of animals and AAIs on human health [19].

School districts across many communities have begun considering or utilizing AAIs, such as therapy dog visits and dog reading programs. However, the literature investigating AAI-related perceptions among educational professionals or school staff is limited. When it comes to perceptions regarding human–animal interactions or AAIs in schools, the data is non-existent in communities such as San Antonio, TX. Previous qualitative studies assessing AAI-related perceptions have mostly been conducted in the context of an evaluation where school professionals may share their perceptions on the effects and outcomes of a specific AAI program. Hence, there is a need to explore other important AAI-related perceptions including perceptions of relative advantage (how much the innovation is perceived to be a better option than what is already in place) and perceptions of compatibility (how much the innovation is compatible with the existing values and needs of those adopting) [20]. To help address this gap in research and contribute to the growing interest in AAIs among San Antonio schools, the current study aimed to explore current knowledge and perceptions relating to animals, AAIs, and their impact on school children among school staff.

## 2. Materials and Methods

### 2.1. Study Design/Approach

A qualitative exploratory study using semi-structured interviews was conducted to explore knowledge and perceptions among school staff as a key group with significant influence on the social-emotional development of school children and the adoption and implementation of school interventions. Specific constructs that were assessed included the following: knowledge of the health impacts of human–animal interactions, perceived benefits of AAIs, relative advantage of AAIs, and perceived compatibility and support of AAIs in schools. A semi-structured interview guide was utilized to successfully conduct each interview. Previous research assessing AAI-related knowledge, perceptions, and/or attitudes among various populations was reviewed to provide insight in the development of questions [21,22,23,24,25].

Finalized questions to assess each construct included, “In what ways can interactions with animals or pets affect a child’s health and well-being?” (knowledge), “In what ways do you think animal-assisted interventions could impact students, particularly in preschool and elementary school grades?” (perceived benefits), “In what ways would the use of animal-assisted interventions align/not align with your school/school district?” (perceived compatibility), and “How do you think parents might feel about animal-assisted interventions in the school and having their children participate?” (perceived support). In the interest of AAIs as a potential strategy to mitigate the effects of trauma, some questions were specific to children who have experienced trauma or behavioral issues (e.g., “In what ways would these interventions be more, or less, beneficial to trauma-exposed students than other school programs or activities you are currently implementing?”). Prior to asking AAI-related questions, participants were provided with a definition of AAIs and a few examples of AAIs. This helped ensure that all participants, regardless of their previous exposure to AAIs, had a basic understanding of this term.

### 2.2. Sample Recruitment and Data Collection

Given the existing literature supporting positive outcomes among young school children and the influence of school staff in the adoption of school innovations, we focused on recruiting school staff that were serving preschool and elementary school children (Prekindergarten through 2nd grade). With previously established relationships between the lead author and school administrators from two San Antonio school districts, purposeful sampling was utilized to recruit staff from specific campuses within these school districts. We coordinated to have administrators send out an email to staff members, which included a brief description of the study and instructions for staff to contact us if interested in participating. We followed up with those who expressed interest to provide a study letter of information and to schedule their interview. Verbal consent was confirmed with each participant during their scheduled interview prior to beginning the first question. A second round of emails was sent prior to the end of the semester as a reminder to those who had not yet responded. Given the impact of the COVID-19 pandemic and the need for continued social distancing, interviews were conducted via Zoom and audio-recorded. All interviews were conducted by the lead author and varied in length (no more than one hour). Upon completing interviews, participants were emailed a USD 20 electronic gift card. These procedures were reviewed and approved by the UTHealth School of Public Health’s Committee for the Protection of Human Subjects (CPHS), as well as administration within each school district (principals, superintendents, and research office).

### 2.3. Data Analysis

Braun and Clark’s six-step framework for thematic analysis was utilized to analyze the data [26]. The first step consisted of the PI listening to interview audio recordings, reading and re-reading transcripts, as well as reviewing any notes recorded during the interviews. In the second step the lead author began organizing data into “chunks” or segments with meaning and assigning them codes. Each coded segment captured something of interest about a specific construct (e.g., knowledge, perceived benefits, etc.). Codes were modified throughout the process as new data were being analyzed. In the third step, codes were grouped together to form themes. In the fourth step, the lead author reviewed themes to ensure they were appropriate, clear, and distinct from one another. In step five, all themes and associated codes were reviewed and discussed with the study team to finalize. In this step, the lead author met virtually with the study team members to present visual maps depicting all themes connected to their associated codes and sample quotes. Upon reviewing and discussing these maps, the study team provided feedback and helped modify theme titles to ensure there would be an accurate representation for each group of codes. In the final step, findings were reported with specific quotes being presented to represent each respective theme.

## 3. Results

A total of 11 school staff members serving preschool and elementary school age children participated in the study and completed a virtual interview. Our approach to saturation in this study follows the data saturation model, which focuses on the redundancy of information as an indicator of saturation [27]. In the last interviews conducted, perceptions and ideas that had previously been discussed were resurfacing and being repeated, which was an indication that we had reached (or were close to reaching) saturation. While the intent was to recruit additional participants to confirm saturation, we reached a point of no response during recruitment.

Participating staff included a social worker, behavior specialist, family and community education specialist, district truancy officer and clerk, two counselors, and three teachers. The number of years in their occupation ranged from one year to nine years. While staff roles were diverse, all participants had experience serving children and families across different grade levels, including those with behavioral or social-emotional issues. One staff member stated she was certified in animal-assisted therapy while others referenced experiences where they had animals visit their school or classroom. While participants were provided with a definition and examples of AAIs, it was not clear whether these visits were formally part of an AAI.

Upon coding and analyzing the data, four key themes were found to be prominent. These themes were “Animals/AAIs supporting emotional well-being”, “Animals/AAIs fostering social-emotional skills”, “Animals/AAIs supporting school values and culture”, and “Concern over children’s safety and well-being when interacting with animals”.

### 3.1. Animals/AAIs Supporting Emotional Well-Being

When participants were asked to share how they thought interactions with animals could impact children’s health and well-being, many were quick to share their own past experiences. These experiences highlighted how interactions with animals led to positive effects on their feelings and emotions. Participants specifically described decreases in negative emotions, such as anger and sadness, and decreases in feelings of anxiety and stress. Participants also mentioned increases in positive emotions such as happiness, joy, and excitement when interacting with an animal. A social emotional counselor remembered her own experience as a child and stated,

“I had dogs, cats, fish. I’d watch fish swim in an aquarium and that would just kind of help mellow things out if I ever felt anxious and um, you know, petting an animal, they say reduces stress by 20% within a minute. So, I would just like, you know, pet my cat or my dog.”

To further explain why they felt animals could support emotional well-being, participants highlighted two attributes specific to animals—their non-judgmental nature and their “calming effect.” This calming effect was often associated with the sensory experience of touching a soft animal. A behavior specialist stated,

“I’ve always found that animals can help be a calming influence, um, for children and adults alike. And even the sensory experience of feeling an animal, feeling the heat of an animal, the soft fur, or, um, even just the pressure of them up next to you or next to your hand or on your hand, um, can be a kind of regulating experience for children.”

Participants believed that an animal’s non-judgmental nature was connected to their inability to talk or react negatively to what a human is saying. In discussing how this could benefit children, a teacher explained, “they’re not gonna get that reaction…either having someone shut them down or tell them anything back. So it’s kind of a different relationship you see with them.” While discussions were not geared toward a specific type of animal, participants often referenced dogs as they brought up past experiences.

Both these attributes were re-emphasized when participants discussed their perceptions of AAIs in comparison to other school interventions or programs that aim to support mental health, particularly among at-risk or trauma-exposed students. Participants believed the involvement of animals was a less threatening approach when working with children who have experienced trauma and may be struggling to feel safe or to open up with an adult. A social worker further explained,

“It goes back to safety, and when someone sees, or a little one sees an animal, it brings that sense of calmness and safety…sometimes you won’t be able to talk to them about a traumatic event, but they feel comfortable and on that little dog’s level or the little cat, um, and be able to have open conversations with ‘em. And sometimes that leads to helping, you know, investigators or teachers or whoever it is kind of identify what they’re going through.”

Multiple participants perceived this calming effect and inability to pass judgment as an added benefit to AAIs that other school programs or interventions may not be able to duplicate.

This theme ultimately highlighted participants’ positive perceptions of animals and AAIs in schools in relation to the perceived impact on children’s emotional health. The mechanisms by which they believed animals could affect emotions included the physiological response (“calming effect”) they believed children could experience when interacting with or touching an animal, as well as the judgement-free space animals could provide. While this theme focused more on perceptions of the internal benefits to children, participants further linked these perceived benefits to other observable outcomes. Discussions around these outcomes became the basis of the second theme.

### 3.2. Animals/AAIs Fostering Social-Emotional Skills

While the term “social-emotional skills” was not always used, most participants described these types of skills as they expressed how they felt having animals and AAIs in the school could potentially benefit children. There are numerous skills or competencies that can fall under the social-emotional category; however, participants specifically highlighted empathy, the ability to manage or control negative emotions, as well as social engagement or relationship-building in the classroom.

Participants used the term “empathy” multiple times in their responses. Interactions with animals were specifically perceived as an opportunity to learn or better understand how others feel. A counselor who discussed this concept further explained,

“I think of empathy a lot when it comes to animals. Um, like teaching children or people just how to…um, to be calm, to be tender, just to be empathetic to what the animal is feeling and then to pick up on cues that way. Okay, like you see how the dog is, you know, kind of backing away from you, why do you think that is? Or—hey, the dog’s tail is wagging, why do you think that is?”

While animals involved in AAIs have their handlers to oversee their care, participants felt children would play an active role in caring for these animals. In having shared responsibility for the animal’s well-being, participants felt children would be able to practice empathy and ultimately apply this skill in their interactions with peers. This perceived learning opportunity was not described in connection to any specific type of AAI, but rather to the general experience of being able to interact with an animal. When asked to compare to other interventions or programs with similar aims, participants perceived animals as having the ability to capture children’s attention with much more ease than other activities or curricula.

Participants also believed that AAIs and interactions with animals could help foster or improve peer-to-peer engagement as well as the management of negative emotions (emotional regulation). This perceived improvement in emotional regulation was associated with the calming effect that was initially described in the emotional well-being theme. A behavior specialist explained, “if one of my kiddos is upset, they sometimes will go to one of our pets and…and cuddle and use that as a self-regulating tool to kinda get back down to calm. And we’ve definitely encouraged that”. Personal experiences were also shared to convey the perceived impact on peer-to-peer engagement. One teacher recalled an experience involving two children that were having issues socializing with their classroom peers. She explained,

“…there was not much interaction between them and other students. And so we noticed when they were put with other students and had the puppies there, their engagement with each other, with, you know, a conversation…we saw a lot of social skills that we saw were lacking in the classroom.”

While participants did not discuss in depth how they thought peer-to-peer engagement would be improved with an animal present (other than it sparking more verbal conversations among peers), it was perceived to be a positive step towards building relationships and building community.

This theme ultimately highlighted participants’ positive perceptions of animals and AAIs in schools in relation to the perceived impact on social-emotional skills. Those with previous exposure to animals on their school campus shared their personal experiences to reinforce the notion that animals and AAIs could potentially provide a unique learning experience. This theme made it evident that interventions that support social-emotional development are of important value in schools. As more emphasis was made on school values and culture, a new theme emerged.

### 3.3. Animals/AAIs Supporting School Culture and Values

All participants, whether they had experience with AAIs or not, felt that AAIs would support and align with current school values and culture. The concept of being able to serve the whole child was stressed and discussed by multiple participants as an important part of culture that would make AAIs appealing or fitting in the school community. A counselor further explained, “right now in the world of education, the push is to serve the whole child. So, um, the philosophy is that you cannot educate academically, educate an individual if you leave out the social and emotional component.” This whole child concept described by participants also included supporting mental health. Furthermore, there was consensus that AAIs complemented current school efforts or programs, such as social-emotional learning (SEL) and the “Handle with Care” program. A few participants pointed out how their school curriculum and activities were centered on student interests and providing different options for learning. A teacher stated, “our school is very based on student interest, um, so maybe seeing…what animals the students are interested in for those who are going to use these resources…” Participants felt that children generally have interest in animals (particularly dogs) and perceived the involvement of animals to be an advantage of using AAIs in schools.

When discussing how others in the school community may feel about AAIs and having animals on their school, participants generally thought that these interventions would receive support from other school staff, administrators, and parents. Participants that had already been exposed to some type of animal visit on their school campus expressed that previous support and buy-in had been garnered. A behavior specialist expressed,

“All of us here want to do what’s best for kids...the environment at the schools was very much like, Yes, we wanted it, we wanted the kids to be able to experience it, and have this, you know, these little sessions or this little time with the animals.”

The willingness to try something new that can help children succeed (especially those with trauma) seemed to be a norm in school programming, and participants felt that education on AAIs and how they work would yield even better support for the adoption of these interventions.

This theme ultimately highlighted participants’ positive perceptions of AAIs and ani animals in schools in relation to the perceived alignment they have with school values and culture. These perceptions made it evident that academics were viewed as just one piece of the puzzle when it comes to serving school children. The value placed on mental, social, and emotional well-being was emphasized, and the need to address students’ needs in these areas was conveyed as important work that AAIs could potentially support in the school. In addition, AAIs were seen as interventions that could cater to students’ interests and that would be supported by others in the school community. However, the perceived support that participants expressed would not be absent of perceived concerns or challenges. As participants discussed some of the issues they felt may arise in the adoption of AAIs, a new theme emerged.

### 3.4. Concern over Children’s Safety and Well-Being When Interacting with Animals

Participants felt that some school children, parents, and/or teachers may be fearful of being bitten or of having an allergic reaction. The root of some of these fears were perceived to be related to limited exposure to animals or past negative experiences involving animals. A behavior specialist further explained,

“Let’s say your whole life you’ve been raised with a cat. And then a dog just comes bounding up to you. And it’s, you know...our brain automatically puts us into, you know, fight, flight, or freeze because it’s something different and it doesn’t fit the pattern or it doesn’t…it’s outside of what we, um, are used to experiencing.”

Personal experiences were also shared in which participants had witnessed school parents or children responding in a fearful way to animals, especially dogs. A truancy clerk mentioned, “We have the little ones, sometimes a dog comes near the playground and they’re screaming and yelling because there’s a dog outside.” Some of the experiences shared were more specific to a type of animal or an attribute (e.g., larger dogs), suggesting that concerns relating to safety may not be tied to all animals and/or may stem from stereotypes or misinformation. When it came to allergies, participants mainly emphasized the importance of having proper planning and established protocols. A district truancy officer stressed,

“We would have to make sure to get, uh, release from every student…or know if any student is allergic to that because we don’t wanna be held responsible for, uh…or…or put a child in danger of having some sort of attack.”

This theme ultimately highlighted some key concerns when it comes to having AAIs and animals in a school setting. Contributing factors highlighted by participants included past trauma involving an animal, unfamiliarity or misinformation about certain animals or breeds, as well as allergic reactions. While some participants felt that school administrators may see these concerns as liability issues, they expressed potential solutions or strategies they felt would ultimately help address these concerns. These solutions included, but were not limited to, offering alternate activities when AAI are implemented, allowing parents and children to observe interactions with animals first before consenting to participate, and using different animals to which children may not be allergic or fearful.

## 4. Discussion

Findings from this study revealed that school staff held positive perceptions towards having animals in a school setting and the use of AAIs. Staff also had some knowledge of the potential benefits of human–animal interactions that are found in the literature. Those who had previously observed or experienced having an animal on their school campus used their experience to further explain why they held these perceptions. All final themes support existing literature on the human–animal bond, its impact on mental health, as well as the social-emotional benefits among participating children [9,28]. Findings also support previous research investigating potential challenges or concerns relating to specific AAIs. Previous studies with different populations have found fear of animals and allergic reactions to be some concerns (among many others) when it comes to having animals in school settings [25,29,30]. Despite such concerns, other evaluative studies considering specific AAI programs have found positive perceptions among various school populations [21,31,32,33].

### 4.1. Limitations

Due to lengthy approval processes and restricted access during the COVID-19 pandemic, our recruitment was limited to select campuses in two school districts. Despite efforts to recruit additional staff from participating schools, workloads continued to increase as efforts to mitigate the effects of the pandemic on student learning became a primary focus. Funding was also limited, impacting the ability to expand incentives to recruit additional schools and school districts. Most participants were female staff and no demographic information beyond gender and position title were collected for this study. Hence, these findings are preliminary and are not generalizable to the larger San Antonio community nor to all school staff/professionals. We further recognize that the term “animal-assisted interventions” was used broadly without focusing on a specific type of AAI or type of animal (e.g., animal-assisted therapy vs. humane education, dogs vs. horses, etc.). It is possible that knowledge and perceptions may vary according to the type of AAI or the type of animal(s) that are being considered. Additionally, we did not address the importance and implications of animal welfare when considering AAIs in schools. The limited number of AAI studies that have included aspects of animal welfare highlights the need to better consider this critical issue and work towards establishing a comprehensive and mutually beneficial “human–animal approach” for implementing and evaluating AAIs [34]. Finally, we acknowledge the potential for interview bias. When asking participants to think about AAIs in comparison to other school interventions, there was no probing into what these other interventions entail and how staff may have evaluated such interventions to further understand why participants may view AAIs as more beneficial.

### 4.2. Implications for Educators and Administrators

School districts seeking to implement or expand AAIs may benefit from engaging key stakeholders, such as school staff, and considering their perceptions or concerns early in the implementation process. Schools with limited experience or exposure to AAIs may find value in others’ experiences and perceptions in relation to interactions with animals and AAI implementation. Such information can assist with early planning and the development of strategies to address potential issues or challenges. While potential solutions to common concerns were mentioned by study participants, schools should consider evaluative research and approaches that have been beneficial in other communities. For example, schools may consider using hypoallergenic dogs, develop proper disinfecting and handwashing protocols, and allow observations [35]. These considerations can ultimately help educators and administrators develop appropriate policies for the implementation of AAIs in school settings.

### 4.3. Future Research

Further research with a larger and more diverse sample is needed to assess existing knowledge relating to the impact of AAIs on children’s mental or emotional health, as well as perceptions relating to the use of AAIs in schools. Beyond the context of the San Antonio community, these preliminary findings provide a segue for expanding AAI-related research, not only to evaluate specific school-based AAI programs, but to help assess whether such interventions are perceived to be compatible with school values and culture and how they are perceived in comparison to other interventions. While those with substantial experience in AAIs can offer deeper insight into implementation, those with less knowledge and experience can also provide insight into potential misconceptions or concerns that can affect the adoption of such interventions. Future attempts to understand AAI-related perceptions in this population should consider potential demographic differences. Furthermore, it would be beneficial to carefully consider perceived challenges in the implementation of AAIs in schools and better understand how educators and administrators would consider the welfare of participating animals in addition to other safety concerns.

## 5. Conclusions

Expanding efforts to support mental health among children has become a priority for many communities, where schools and other youth-serving organizations are striving to mitigate the effects of trauma. The use of animal-assisted interventions (AAIs), which have been shown to have positive outcomes in educational settings, is continuing to expand in communities such as San Antonio. Positive perceptions of AAIs, as well as existing knowledge of the potential health benefits of human–animal interactions, can be influential in expanding the adoption and implementation of these interventions. Future research will help us better understand the perceived impact of human–animal interactions and AAIs among school stakeholders, as well as the potential challenges involved in implementing these interventions in school settings. This information can ultimately help schools make decisions on the adoption and implementation of AAIs and shape future policies.

## Data Availability

The data presented in this study are available on request from the corresponding author. The data are not publicly available due to privacy.

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
