# Peer review of "Exploring School Staff Perceptions Relating to Animals and Their Involvement in Interventions to Support Mental Health"

_ijerph, 2022, doi:10.3390/ijerph19127126_

Round 1

Reviewer 1 Report

Introduction is clear and well-structured with a good rationale for conducting the study. Minor comment – please rephrase ‘use of animals’ to ‘involving animals’, as animals are not tools within interventions (e.g., line 46, line 174, line 210).

Very clear methods section including all relevant detail.

Results are very interesting. The fact the authors include potential limitations of AAIs (within the fourth theme) really strengthens the paper, as research has often promoted the blanket message that AAIs are beneficial for everyone. This is an important message the authors have highlighted and a particularly valuable contribution to the field.

I appreciate the authors acknowledge that a larger and more diverse sample is required in the future, but I think it is important to emphasise that only 11 participants were included so these findings are important yet very preliminary and cannot be generalised in any way as it is such a small sample size (understandably so given the design and setting, but I think it is worth highlighting in the discussion).   

Overall, I really enjoyed reading the paper and thank the authors for their valuable contribution.

Author Response

Point 1: "Please rephrase ‘use of animals’ to ‘involving animals’, as animals are not tools within interventions (e.g., line 46, line 174, line 210)."

Response 1: Previous terms have been rephrased to read “involve animals” and “involvement of animals” (see revised manuscript).

Point 2: "I think it is important to emphasize that only 11 participants were included so these findings are important yet very preliminary and cannot be generalized in any way as it is such a small sample size (understandably so given the design and setting, but I think it is worth highlighting in the discussion)."

Response 2: The following content has been added to the discussion/limitations section (starting on line 362) – “Due to lengthy approval processes and restricted access during the COVID-19 pandemic, our recruitment was limited to select campuses in two school districts. Despite efforts to recruit additional staff from participating schools, workloads continued to in-crease as efforts to mitigate the effects of the pandemic on student learning became a primary focus. Funding was also limited, impacting the ability to expand incentives to recruit additional schools and school districts...(line 369) Hence, these findings are preliminary and are not generalizable" (see revised manuscript).

Reviewer 2 Report

This is such an important topic and addresses a gap in the literature. Some minor suggestions to strengthen the manuscript.

-Modifying the title so the reader knows that animal interactions were not implemented at the school. 

-In your abstract/introduction, clarify the age group you are talking about. 

-Include demographic information about your participants as well as how many out of the 11 had previous experience with AAIs.

-Include some additional information about the trustworthiness of your study (i.e., member checking, more details about the meeting with "study team"-peer debriefing) 

-Funding was not mentioned as a potential challenge 

- Your discussion section needs a little content. Possibly break it into subsections (discussion, implications for educators/administrators, limitations, future research). Since this is an exploratory study you want to set yourself up for your next study. 

Author Response

Point 1: Modifying the title so the reader knows that animal interactions were not implemented at the school.

Response 1: The study title has been revised to – Exploring School Staff Perceptions Relating to Animals and their Involvement in Interventions to Support Mental Health

Point 2: In your abstract/introduction, clarify the age group you are talking about.

Response 2: The following statement has been included in the abstract (starting on line 12) – “A total of 11 interviews were completed with staff serving preschool and elementary school age children.” This clarification has also been made in the results section (see revised manuscript).

Point 3: Include demographic information about your participants as well as how many out of the 11 had previous experience with AAIs.

Response 3: No demographic information was collected from participants other than their position title and number of years in their role. The question “Have you been previously involved in an AAI?” was not formally asked. However, in the results section we have clarified and made the following statement (starting on line 162) – “One staff member stated she was certified in animal-assisted therapy while others referenced experiences where they had animals visit their school or classroom. Whether these visits were formally part of an AAI was not clear” (see revised manuscript).

Point 4: Include some additional information about the trustworthiness of your study (i.e., member checking, more details about the meeting with "study team"-peer debriefing)

Response 4: We have added the following content to provide more insight into the process of finalizing themes with the study team (starting on line 148) – “In this step the lead author met virtually with study team members to present visual maps depicting all themes connected to their associated codes and sample quotes. Upon reviewing and discussing these maps, the study team provided feedback and helped modify theme titles to ensure there would be an accurate representation for each group of codes” (see revised manuscript).

Point 5: Funding was not mentioned as a potential challenge

Response 5: The following statement has been added to the limitations section (starting on line 366) – “Funding was also limited, impacting the ability to expand incentives to recruit additional schools and school districts” (see revised manuscript).

Point 6: Your discussion section needs a little content. Possibly break it into subsections (discussion, implications for educators/administrators, limitations, future research). Since this is an exploratory study, you want to set yourself up for your next study. 

Response 6: The following subsections were created – Limitations, Implications for Educators & Administrators, and Future Research. Content within these subsections were also updated/expanded (see revised manuscript).

Reviewer 3 Report

Thank you for the opportunity to read this interesting paper that explores the views of educational professionals relating to the potential benefits they can imagine from embedding AAI into schools. It is a well written manuscript with a solid qualitative methodology. I have some comments, which though longwinded, can be summarized into 4 themes: 1) Firstly, I think more literature – and particularly, more contemporary studies of AAIs, would help the authors refine their contribution and connect to existing conversations. 2) Secondly, I think there is an opportunity to tweak the research question slightly, highlighting the focus of working with educational professionals as a specific group, rather than leading with the geographic context as the ‘unique selling point’ of this paper. 3) Thirdly, whilst I thought the methods were good, I do think we need a little more detail on why only 11 interviewees/how saturation was reached. And 4) fourthly, a slight tempering of language and how the qualitative data is presented, to remember that this is people talking about their perceptions, beliefs, anticipations, imaginations, etc. etc. of AAI, this is there at times, but also sometimes shifts to presenting these views as ‘facts’ relating to efficacy.

The introduction is very brief, only skimming the existing literature on AAIs and AAI/Trauma links very quickly. I think more detail would be helpful here, this is an area that has developed a rich literature recently. Additionally, after checking the reference list, references 8 and 9 are websites, rather than academic studies. This is not really appropriate for a journal article, again, particularly given there are whole journals dedicated to exploring the topics of AAI – I suggest the authors check out articles in Anthrozoos or Society and Animals. Reference 10, similarly, is over 33 years old. Given the burgeoning interest in this topic, this seems a shame and signals a lack of engagement with more recent research.

The paper feels very ‘local’, in that it appears to be making a case for a gap in research based on lack of knowledge about perceptions regarding AAIs in one particular US city (line 59-62). Considerations need to be given to the relevance and generalizability elsewhere, and also, why existing literature on perceptions of AAIs are not applicable here? Is this really a gap? What makes San Antonio unique that existing literature might not apply in this context? I certainly think this is a valuable study, but I think more careful thought needs to be given to how the central ‘hook’ of novelty and uniqueness is sold to the reader – that no one has studied perceptions of AAI in this particularly city isn’t quite enough. The uniqueness of the research question should ideally be independent of the context.

The methodology is well written, and the authors should be commended for their detailed description of their analytic approach. My only methodological question would be how the authors decided when they had reached a level of ‘saturation’ – 11 interviews, with some only lasting 30 minutes, seems a very limited sample, and I wondered why further research was not carried out. A quick google suggests there are 17 school districts in San Antonio, yet only staff from two districts were invited and participated. There is a question of how generalizable this can be to the whole of ‘San Antonio school staff’ as is identified in the research question on line 70. Given the specificity of San Antonio is currently quite fundamental to the article’s claims around uniqueness and novelty, this is quite a challenge, and again I think a prompt to slightly reframe the question/focus to perhaps be more about professionals, independent of a specific geographic context.

There is a tension around what the qualitative data is being used to claim. We must remember that these are perceptions of benefit, rather than making the claim that animals DO support wellbeing. It may requiring rethinking about how themes/subheadings are titled and presented. This is particularly important given that some of the interviewees didn’t seem to actually have experience of AAI (line 135-138)? Or is more clarity needed here? As otherwise, it’s perhaps a strange choice to interview people about a topic (AAI) they have no experience of, and limits what can be taken from their claims – beyond thinking about what they might ‘imagine’ the benefits of AAI to be (a subtle difference from claims about what the benefits of AAI are). For example, several of the quotes (line 153) talk about people’s previous experience of pet ownership. This is very different to formalized AAI, and shouldn’t really be used to make claims about the efficacy of AAI. People are talking in quite vague and anecdotal terms (line 161) rather than specific to AAI. Really, 3.1 is talking more about ‘why professionals might be interested, or have a belief in, AAI being useful’.

A little more critical thought needs to be put in to thinking about what the qualitative data is being used to claim, and some tempering of language introduced (using terms like perception, belief, anticipation…). More reflection is needed as to who was interviewed, and why, and their ability, authority, and veracity to be able to make claims about AAI interrogated a little bit further.  Section 3.3 is written much better, recognizing straight away that we are talking about perceptions and feelings here, and acknowledging (line 269) that interviewees without exposure to AAIs might have reasons for different perspectives.

I think the interesting thing to pull out here might be WHY staff believe these things about AAI, for example, if we take lines 215-216, where ‘participants acknowledged that animals can most likely capture children’s attention with much more ease than other activities or curricula’ – firstly, this is less an acknowledgement (which feels like a biased term), and certainly a perception/belief. But the key question is, why do staff believe that, how have they come to that conclusion, what does it tell us about the attraction of AAI. Indeed, what even are these ‘other interventions’ vaguely described? How have staff evaluated and arrived at that decision? Is there a risk of interview bias? There is a risk of just reporting and describing participants views, rather than critically engaging with them or analyzing them.

The conclusion is very short and currently doesn’t say much. What is the impact of this study beyond San Antonio? Why this journal? The conclusion also suggests that positive perceptions of AAI demonstrate a potential to expand implementation – but this seems too simplistic, surely there are many other challenges? I think a more thorough engagement with recent literature on AAI would be helpful at the start and again here, and allowing the authors to showcase how their research is contributing to current conversations about AAIs. Trauma, which seemed an important focus at the start, has sorted of faded away by the conclusion too?

There is also a risk of instrumentalizing animals. Animal welfare is not mentioned, nor the impact on the animals involved in AAI. Much recent AAI research now takes the approach that these things need to be considered as much as participants safety and wellbeing. There are ethical considerations involved in thinking about AAI, yet the article calls with a broad call for expansion of AAIs. Again, there are opportunities to be a little more critical and recognize some of the challenges, limitations, and complexities involved, rather than a wholehearted enthusiasm.

Author Response

Point 1: “I think more literature – and particularly, more contemporary studies of AAIs, would help the authors refine their contribution and connect to existing conversations.”

Response 1: An additional paragraph (starting on line 61) has been included to highlight more recent AAI studies (within the last 5 years) specific to educational settings and/or relating to mental health (see revised manuscript).

Point 2: “I think there is an opportunity to tweak the research question slightly, highlighting the focus of working with educational professionals as a specific group, rather than leading with the geographic context as the ‘unique selling point’ of this paper.”

Response 2: To better establish the uniqueness of this paper, we have added content to highlight how previous studies investigating AAI-related perceptions among educational professionals have been limited. Previous qualitative studies have mostly been done in the context of an evaluation where staff share their perceptions on the effects and outcomes of a specific AAI program (this study was not bound by a specific AAI program and allowed participants the opportunity to discuss knowledge and perceptions regardless of their involvement with a specific AAI). This information has added to the last paragraph of the Introduction section (starting on line 79) and the discussion (starting on line 355). The abstract has also been modified to help shift some of the focus away from the geographic context (see revised manuscript).

Point 3: “I do think we need a little more detail on why only 11 interviewees/how saturation was reached.”

Response 3: With the COVID-19 pandemic, access to school staff became more challenging. Despite efforts to recruit additional staff from participating schools, workloads continued to increase as efforts to mitigate the effects of the pandemic on student learning became a primary focus. This challenge also served as an added barrier to reaching other school districts who already had lengthy approval processes for research requests. Additionally, funding was limited impacting the ability to expand incentives to recruit additional schools and school districts. The limitations section (starting on line 362) has been updated to include this information (see revised manuscript).

Point 4: “A slight tempering of language and how the qualitative data is presented, to remember that this is people talking about their perceptions, beliefs, anticipations, imaginations, etc. etc. of AAI, this is there at times, but also sometimes shifts to presenting these views as ‘facts’ relating to efficacy.”

Response 4: Language in the manuscript has been modified to better indicate that our findings don’t represent facts but rather perceptions, beliefs, etc. relating to animals (specifically interactions with them), AAIs, and the potential impact of such interventions to support school children (see revised manuscript).

Point 5: “If we take lines 215-216, where ‘participants acknowledged that animals can most likely capture children’s attention with much more ease than other activities or curricula’ – firstly, this is less an acknowledgement (which feels like a biased term), and certainly a perception/belief. But the key question is, why do staff believe that, how have they come to that conclusion, what does it tell us about the attraction of AAI.”

Response 5: We have included the potential of interview bias in our limitations section (starting on line 375). When asking participants to think about AAIs in comparison to other school interventions, there was no probing into what these other interventions entail and how staff may have evaluated such interventions to further understand why participants may view AAIs as more beneficial.

Point 6: “What is the impact of this study beyond San Antonio? Why this journal?”

Response 6: Studies relating to AAIs and the human-animal bond have been published in this journal. Furthermore, the following statement has been included in the future research subsection (starting on line 397) – “Beyond the context of the San Antonio community, these preliminary findings provide a segue for expanding AAI-related research, not only to evaluate specific school-based AAI programs, but to help assess whether such interventions are perceived to be compatible with school values and culture and how they are perceived in comparison to other interventions.”

Point 7: “The conclusion also suggests that positive perceptions of AAI demonstrate a potential to expand implementation – but this seems too simplistic, surely there are many other challenges?”

Response 7: The conclusion section has been updated and we have stated in our limitations section (starting on line 373) that perceived challenges or concerns may differ according to the type of AAI or animal(s) involved, which was not explored in this study (see revised manuscript).

Point 8: “Trauma, which seemed an important focus at the start, has sorted of faded away by the conclusion too."

Response 8: We thank the reviewer for this observation. While we had questions in our interview guide that specifically addressed AAIs and trauma-exposed children, responses were not specific to that population but more generally to perceptions of AAIs and their impact on school children. We have made language modifications in our introduction to reflect more of a general focus on children’s mental health, which includes trauma (see revised manuscript).

Round 2

Reviewer 3 Report

The authors have responded well to several of my points, and the manuscript is much stronger now. However, inexplicably, the authors have been quite selective in the comments they have chosen to respond to and engage with, they have chosen to ignore several of the other points I raised, without responding or explaining why they are not to engage with these points. I must admit, I was expecting point by point responses, rather than for the authors to pick and choose which aspects of my feedback they responded to. It is a shame, as their responses and adjustments to the comments they chose to respond to were well done.

Points remaining unanswered:

After checking the reference list, references 8 and 9 are websites, rather than academic studies. This is not really appropriate for a journal article, again, particularly given there are whole journals dedicated to exploring the topics of AAI – I suggest the authors check out articles in Anthrozoos or Society and Animals. Reference 10, similarly, is over 33 years old. Given the burgeoning interest in this topic, this seems a shame and signals a lack of engagement with more recent research.

There are 17 school districts in San Antonio, yet only staff from two districts were invited and participated. There is a question of how generalizable this can be to the whole of ‘San Antonio’

Animal welfare is not mentioned, nor the impact on the animals involved in AAI. Much recent AAI research now takes the approach that these things need to be considered as much as participants safety and wellbeing. There are ethical considerations involved in thinking about AAI, yet the article calls with a broad call for expansion of AAIs. Again, there are opportunities to be a little more critical and recognize some of the challenges, limitations, and complexities involved, rather than a wholehearted enthusiasm.

Points requiring some further reflection:

In terms of responding to the point about saturation, the authors blame Covid. I am sympathetic to this point, and I certainly understand how that was a challenge and limitation, but that does not provide an adequate answer to the specific question of ‘saturation’ - as a specific process in qualitative research.

The authors have also introduced a sentence suggesting that “Whether these visits were formally part of an AAI was not clear.” This is a useful reflection, though requires some further editing in consistency in later parts of the manuscript, where the authors return to talking about people describing AAI – for example, at the start of 3.3 the authors claim “All participants perceived AAIs to be interventions” – yet, they have acknowledged that some people had no experience of AAI. The language of “AAIs or animals in their school” used alternatively seems more appropriate.
